# Adaptation and Validation of the Malay Version of the SAVE-9 Viral Epidemic Anxiety Scale for Healthcare Workers

**DOI:** 10.3390/ijerph191710673

**Published:** 2022-08-27

**Authors:** Fatin Syafiqah Wasimin, Sean Chern Choong Thum, Mathias Wen Leh Tseu, Assis Kamu, Chong Mun Ho, Nicholas Tze Ping Pang, Seockhoon Chung, Walton Wider

**Affiliations:** 1Faculty of Medicine and Health Sciences, Universiti Malaysia Sabah, Kota Kinabalu 88400, Sabah, Malaysia; 2Department of Psychiatry, Universiti Putra Malaysia Teaching Hospital, Serdang 43400, Selangor, Malaysia; 3Faculty of Science and Natural Resources, Universiti Malaysia Sabah, Kota Kinabalu 88400, Sabah, Malaysia; 4Department of Psychiatry, Asan Medical Center, University of Ulsan College of Medicine, Seoul 05505, Korea; 5Faculty of Business and Communications, INTI International University, Nilai 71800, Negeri Sembilan, Malaysia

**Keywords:** viral epidemic, COVID-19, psychological impact, healthcare, healthcare workers, SAVE-9 scale

## Abstract

Viral epidemics have surfaced frequently over the past quarter-century, with multiple manifestations of psychological distress. This study sought to establish the psychometric properties of the Malay version of SAVE-9 among healthcare workers. A total of 203 healthcare workers across Malaysia participated in the research. The Malay version of SAVE-9 was translated and back-translated using the WHO instrument validation protocols. Classical Test Theory (CTT) and Rasch analysis were used to assess the validity and reliability of the Malay version of the SAVE-9 scale. The analysis was run using IBM SPSS 26.0 and JAPS. Cronbach’s alpha was used to measure the internal consistency of SAVE-9, which was found to be satisfactory (Cronbach’s α = 0.795). The correlations between the SAVE-9 and other measured scales (GAD-7 and PHQ-9) were statistically significant. A score of 22 was defined as a cut-off point with good sensitivity (0.578) and specificity (0.165). The Malay version of the Stress and Anxiety to Viral Epidemics-9 (SAVE-9) scale is valid and reliable after testing among healthcare workers. It is psychometrically suitable to be used in assessing healthcare workers’ stress and anxiety specific to viral epidemics.

## 1. Introduction

The first quarter of the 21st century provided many challenges to the world population, especially in the healthcare fraternity. Major viral epidemics and a pandemic have been observed since 2003, when the Severe Acute Respiratory Syndrome (SARS) outbreak in China was brought to attention at an international level, prompting the World Health Organization (WHO) to declare it as a serious global health threat [1]. The world suffered with subsequent emerging outbreaks, such as the H1N1 influenza in 2009, the Middle East Respiratory Syndrome (MERS) in 2012, the Ebola outbreak in 2013 and the SARS-CoV-2 (COVID-19) pandemic.

First detected in December 2019, COVID-19 rapidly spread all over the world in a few weeks leading to the disease being declared a public health emergency of international concern by the end of January 2020 [2]. The healthcare systems all over the world were significantly affected by the sudden increase in workload, demands for medical equipment, congested health facilities and many more. As a result, healthcare workers (HCW) became prone to psychological distress and burnout in their work and their personal lives [3,4,5,6,7]. This has been proven by various studies carried out across the globe. A prevalence of more than 50% was demonstrated for anxiety, insomnia and depression among HCW in China [8], while a study involving eight European countries showed that a considerable proportion of HCW showed high values for depression, anxiety and stress [9]. In Malaysia itself, a cross-sectional study that was conducted revealed that the prevalence and severity of depression, anxiety and stress were maintained at a high level, regardless of the relaxation of lockdown in the country [10]. While these studies incorporated the use of generic psychopathology scales to measure the level of psychological distress, such as the Depression, Anxiety and Stress Scale (DASS-21), the Generalized Anxiety Disorder (GAD-7) scale and Patient Health Questionnaire (PHQ-9), a brand new tool called the Fear of COVID-19 scale (FCV-19S) was developed to specifically assess the psychological impact of COVID-19 in the general population [11]. Its reliability was tested and validated in multiple languages, including the Malay language [12]. However, the unique attributes of this scale covering COVID-19, as the name implies, limits its use in other disease settings. Mental health wellbeing among HCW was also demonstrated to be significantly affected during the previous viral epidemics, particularly in the past two decades. The measured psychiatric morbidity among HCW in a study in China during the SARS epidemic was estimated to be as high as about 75% [13], while the highest risk for post-traumatic-stress disorder symptoms was shown among HCW who performed MERS-related tasks during the outbreak [14].

Hence, in light of the need to systematically assess the psychological wellbeing of HCW at any point of time in response to a viral epidemic, a new rating scale, namely the Stress and Anxiety to Viral Epidemics-9 (SAVE-9) for healthcare workers, was developed [15]. The validation work was completed in multiple languages [16,17,18,19,20]; additionally, different variations of the SAVE-9 scale were developed and validated to suit different groups of the population [21,22]. The Malay version of SAVE-9 scale for healthcare workers was made available on 23 September 2021 [23]. The objective of this study is to demonstrate the validity and reliability of the Malay SAVE-9 through rigorous statistical models, allowing this tool to be used with confidence for the assessment of psychological wellbeing among Malaysian healthcare workers during a viral epidemic in the near future, and for better preparedness to tackle these psychological health needs.

## 2. Methods

### 2.1. Study Design and Participants

The translation was prepared according to standard WHO guidelines. Two independent researchers, one of whom is a content expert who is familiar with viral epidemics and bilingual in both English and Malay, and the other, a language expert who is more familiar with the Malay language, translated the scale from English to Malay. Following that, two different researchers serving as the content and language experts, respectively, blind to the original translation, back-translated the Malay version into English. The two versions were then compared and scrutinized for major inconsistencies, producing a harmonized version. This translated version was pilot tested on 20 Malay-speaking individuals. A final Malay translation was then produced, correcting any unusual turns in phrases and incongruency with the English version.

The validation study was conducted in various healthcare settings across both Peninsular and East Malaysia. The recruitment for respondents was performed through the convenience, snowball-method sampling. An online survey using Google Forms was utilized for the data collection to comply with the implementation of strict standard-operating procedures on social distancing and movement control orders. A targeted sample size ranging from 200 to 300 was initially planned, as it is considered to be a fair to good sample size for the purpose of factor analysis [24].

### 2.2. Ethical Considerations

The ethical approval was obtained from the Universiti Malaysia Sabah Medical Research Ethics Committee (ref. no.: JKEtika 3/21(5)) prior to the commencement of this project. All of the participants provided informed consent.

### 2.3. Research Instruments

Apart from the informed consent built-in together in the form, the online survey comprises of four main sections as follows.

#### 2.3.1. Sociodemographic Questionnaire

A simple questionnaire requests respondents to provide information regarding their working environment and its nature, which involves details of the work position, duration, involvement in COVID-19-related care, healthcare level and demographic setting. While the respondents’ personal identities were kept anonymous, their age range, gender and marital status were otherwise requested for sociodemographic demonstration purposes.

#### 2.3.2. SAVE-9 Scale

The English version [15] and the translated final Malay version were both administered. The original English scale consists of nine items scored on a five-point Likert scale ranging from 0 (never) to 4 (always). A two-factor structure was adopted, i.e., (1) anxiety regarding viral epidemics, and (2) work-related stress associated with viral epidemics. A cut-off score of 22 for the scale ascertained the levels of stress and anxiety in healthcare workers in response to a viral epidemic, warranting clinical attention. Cronbach’s alpha was used to measure the internal consistency of SAVE-9, which was found to be satisfactory (Cronbach’s α = 0.795). Correlations between the SAVE-9 and other measured scales (GAD-7 and PHQ-9) were statistically significant.

#### 2.3.3. GAD-7 Scale

The GAD-7 [25] is a seven-item questionnaire, a widely used self-administered tool for assessing general anxiety. The items are scored on a four-point scale each, ranging from 0 (not at all) to 3 (nearly every day). The GAD-7 has a cut-off score of 10 or higher, which is an indication of the presence of generalized anxiety disorder. On the GAD-7, the cut points of 5, 10 and 15 may also be interpreted as mild, moderate or severe levels of anxiety. The Malay version of GAD-7 was used in this study [26].

#### 2.3.4. PHQ-9 Scale

The PHQ-9 [27] is a self-reported questionnaire consisting of nine items that mainly assess symptoms of depression. Each item is scored on a four-point Likert scale, ranging from 0 (not at all) to 3 (nearly every day). A cut-off point of 0–4 indicates no depressed symptoms, 5–9 mild depressive symptoms, 10–14 moderate depressive symptoms, 15–19 moderate–severe depressive symptoms, and 20–27 severe depressive symptoms in the PHQ-9 severity scale. The Malay version of PHQ-9 was used in this study [28].

### 2.4. Data Analysis

The classical test theory (CTT) was used to check the validity and reliability of the Malay version of the SAVE-9 scale. The methods employed for reliability were internal consistency measure using Cronbach’s alpha, McDonald’s omega, greatest lower bound, and test–retest reliability using the Pearson correlation test (Malay version versus original English version). For validity, convergent validity (versus GAD-7 Malay version and PHQ-9 Malay version) was used. The dimensionality of the SAVE-9 Malay version was explored, using exploratory factor analysis (EFA), which uses principal axis factor as the extraction method and promax as the rotation method. The effectiveness of the SAVE-9 Malay version as a diagnostic instrument for stress and anxiety and the appropriate cut-off point for the total score of the SAVE-9 Malay version were determined, using the receiver operating characteristic (ROC) analysis. Finally, a comparison test using Kruskal–Wallis was conducted to compare the scores of the SAVE-9 Malay version, according to the groups of participants classified using GAD-7 Malay version and PHQ-9 Malay version. The analysis was run using IBM Statistical Package for Social Sciences (SPSS) version 27 (Developed by Norman H. Nie, Dale H. Bent, and C. Hadlai Hull, Chicago, IL, USA) and JASP Version 0.16 (University of Amsterdam, Amsterdam, The Netherlands). The original version (i.e., the English version) and the Malay version of the SAVE-9 are as presented in Table 1.

## 3. Results

### 3.1. Demographic Information

All of the 203 healthcare workers participated in this study (Table 2). Among the participants, 152 (74.9%) were female, and 94 (46.3%) were single. A total of 114 (56.2%) were medical doctors, and 52 (25.6%) were staff nurses. All of the 154 (75.9%) participants responded that they have cared or are still caring for COVID-19 patients. Twenty-five (12.3%) experienced being quarantined, 39 (19.2%) had a past psychiatric history, and 59 (29.1%) were now experiencing depression, anxiety or needed help dealing with their current emotions or moods.

### 3.2. Dimensionality of the SAVE-9 Malay Version

The normality assumption was checked and we observed that the distribution of all of the items of the Malay version of the SAVE-9 was within the normal limit based on skewness and kurtosis for an acceptable limit of range between −2 and +2 (Table 3). The Kaiser–Meyer–Olkin measure verified the sampling adequacy for the factor analysis, as its value is more than 0.5 (0.865). The Bartlett’s test of sphericity (X^2^(36) = 630.435, *p* < 0.001) also confirmed that relationships existed between at least some of the nine items, which means that the correlation structure was adequate for factor analysis. The principle axis factoring confirmed the bi-dimensionality of the SAVE-9 Malay version, as there were two factors extracted. The eigenvalues for the factors 1 and 2 were 2.815 and 1.394, respectively. Factor 1 can explain 31.3% of the variation in the nine items, while Factor 2 is 15.5%. The factor loadings and the Cronbach’s alpha are as shown in Table 4.

### 3.3. Reliability and Validity of the SAVE-9 Malay Version

All of the psychometric measures’ results, as shown in Table 5, confirmed the validity and reliability of the SAVE-9 Malay version, as all of the values passed the suggested cut-off.

### 3.4. Cut-Off Score for the SAVE-9 Malay Version

The ROC graph, as shown in Figure 1, displayed the convex pattern that indicated good discrimination ability. The Area Under the Curve (AUC) showed excellent diagnostic accuracy (AUC 0.750 (95% CI 0.662–0.838); *p* < 0.001). The appropriate cut-off score was determined as 21.5 or 22, with a good sensitivity (0.578) and specificity (0.165).

### 3.5. SAVE-9 Malay Version Scores Based on Anxiety and Depression Levels

The results of the Kruskal–Wallis test showed that the total scores for the SAVE-9 Malay version, the score of Factor 1, and the score of Factor 2 were significantly different among the three groups based on the anxiety levels (GAD-7 Malay version scores of 0, 1–4, and ≥5) and also based on the depression levels (PHQ-9 Malay version scores of 0, 1–9, and ≥10) (see Table 6).

## 4. Discussion

The Malay version of the SAVE-9 shows superior psychometric properties, with reasonable test–retest reliability, convergent validity to two separate measures and internal consistency using three separate measures. This also shows that there are two distinct factors which are distinguishable from each other, which corroborates the original SAVE-9 factor structure, and there is an identifiable cut-off point on the ROC curves with reasonable sensitivity and specificity. With this, the SAVE-9 is a psychometrically suitable tool to be used in the surveillance the stress and anxiety of HCW specific to a pandemic.

The uncertainty and hopelessness related to COVID-19 are likely to contribute to HCW developing stress and anxiety specific to the COVID-19 pandemic [29]. This is corroborated by studies with evidence that has shown that the HCW in direct contact with COVID-19 patients have higher anxiety levels [10,30,31,32]. This may be attributed to a fear of contamination with the virus, spreading the virus to their family and friends, as well as being responsible for the subsequent morbidity and mortality [33]. Moving on, HCW may be prone to vicarious traumatization, which refers to the negative changes they develop following exposure to other people’s trauma [34,35]. This is because HCW care for patients who are suffering from the physical and psychological “trauma” of COVID-19, which causes increased distress [36]. It is not limited to frontline HCW, but also includes non-frontline HCW [37]. Hence, it is important to be aware of this phenomenon and take proactive measures to prevent long-term consequences for HCW [38]. There are ultra-brief psychological interventions that can be performed in ward settings, which have been demonstrated to be efficacious in providing the quickest form of psychological interventions to reduce morbidity [39]. Telehealth single-session mindfulness interventions have been demonstrated to be equally helpful in improving psychological flexibility and cognitive flexibility [40].

Several limitations were present in this research. There were no data measuring the stress and anxiety levels of HCW during viral epidemics prior to COVID-19, hence a like-for-like comparison could not be made. Due to strict national guidelines, the project had to be conducted online, which may limit participant reach. Furthermore, HCW are involved with the COVID-19 response, so random sampling was a challenge. In order to obtain data, convenience snowball-sampling was the most ideal way to access the respondents. However, this meant the respondents may not be wholly representative of the targeted population. Lastly, as this was a self-reported project aiming to obtain respondents’ points of view, self-reporting bias cannot be avoided. Further qualitative studies can be carried out to investigate the congruency of responses.

### Conclusions

The Malay version of the Stress and Anxiety to Viral Epidemics-9 (SAVE-9) scale is valid and reliable after being tested among HCW. It is thus psychometrically suitable to be used in assessing healthcare workers’ stress and anxiety specific to pandemics. This instrument will be a valuable tool in screening individuals in healthcare settings who have a high risk of pandemic-specific anxieties, allowing expedited referrals to relevant interventions.

## Figures and Tables

**Figure 1 ijerph-19-10673-f001:**
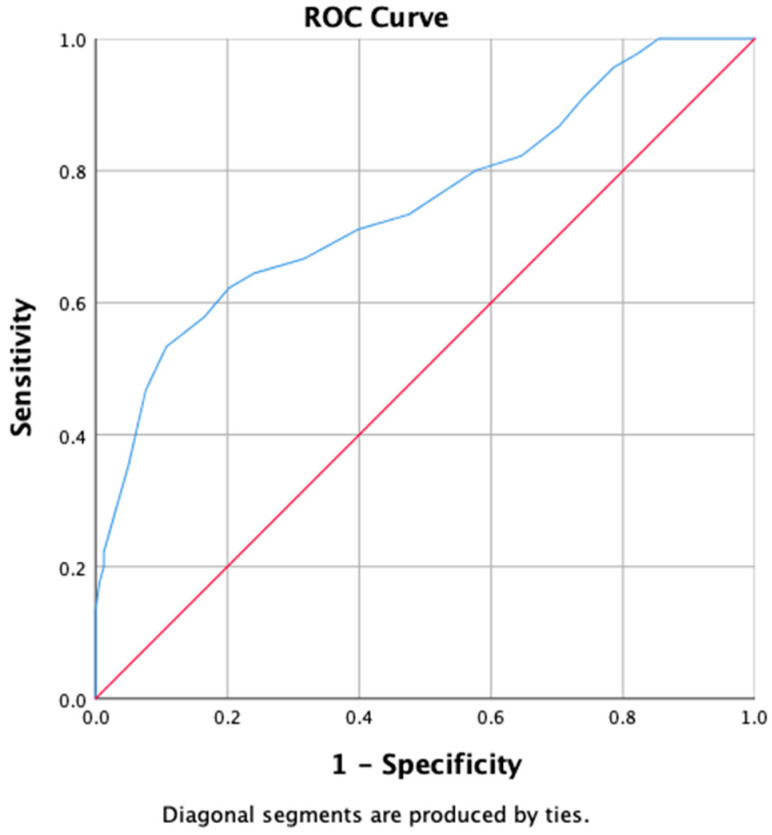
ROC curve sensitivity and specificity of SAVE-9 Malay version compared with GAD-7 Malay version.

**Table 1 ijerph-19-10673-t001:** The original English version and the Malay version of the SAVE-9.

Item	The Original English Version	The Malay Version
Item 1	Are you afraid the virus outbreak will continue indefinitely?	*Adakah anda takut wabak virus akan berterusan selama-lamanya?*
Item 2	Are you afraid your health will worsen because of the virus?	*Adakah anda takut kesihatan anda akan bertambah teruk kerana virus?*
Item 3	Are you worried that you might get infected?	*Adakah anda bimbang anda mungkin dijangkiti?*
Item 4	Are you more sensitive towards minor physical symptoms than usual?	*Adakah anda lebih peka terhadap gejala fizikal yang kecil daripada biasa?*
Item 5	Are you worried that others might avoid you even after the infection risk has been minimized?	*Adakah anda bimbang bahawa orang lain mungkin mengelakkan anda, walaupun selepas risiko jangkitan telah berkurang?*
Item 6	Do you feel skeptical about your job after going through this experience?	*Adakah anda merasa ragu-ragu dengan pekerjaan anda setelah melalui pengalaman ini?*
Item 7	After this experience, do you think you will avoid treating patients with viral illnesses?	*Selepas pengalaman ini, adakah anda fikir anda akan mengelakkan diri daripada merawat pesakit yang disebabkan oleh virus?*
Item 8	Do you worry your family or friends may become infected because of you?	*Adakah anda bimbang keluarga atau rakan anda mungkin dijangkiti disebabkan anda?*
Item 9	Do you think that your colleagues would have more work to do due to your absence from a possible quarantine and might blame you?	*Adakah anda fikir rakan sekerja anda mungkin perlu buat lebih banyak kerja kerana ketiadaan anda disebabkan kemungkinan kuarantin dan mungkin akan menyalahkan anda?*

**Table 2 ijerph-19-10673-t002:** The respondents’ background information (*n* = 203).

Variables		*N* (%), Mean (SD)
Age group (years)	20–29	83 (40.9%)
	30–39	108 (53.2%)
	≥40	12 (5.9%)
Sex	Female	152 (74.9%)
	Male	51 (25.1%)
Marital status	Single	94 (46.3%)
	Married	109 (53.7%)
Healthcare worker	Medical doctor	114 (56.2%)
	Staff nurse	52 (25.6%)
	Other healthcare worker	37 (18.2%)
Years of employment		10 (10.70)
Have you cared or are you still caring for COVID-19 patients?	Yes	154 (75.9%)
	No	49 (24.1%)
Have you ever been infected with COVID-19 disease and underwent a quarantine process?	Never	178 (87.7%)
	Yes	25 (12.3%)
Have you ever experienced or been treated for depression, anxiety, or insomnia?	Never	164 (80.8%)
	Yes	39 (19.2%)
Do you feel that you are experiencing depression, anxiety, or need help dealing with your current emotions/moods?	No	144 (70.9%)
	Yes	59 (29.1%)

**Table 3 ijerph-19-10673-t003:** Descriptive statistics of the Malay version of the SAVE-9 scale (*n* = 203).

Item	Mean	Std. Deviation	Skewness	Kurtosis	Coefficient of Variation	Factor 1	Factor 2
Item 1	2.18	0.95	0.018	0.046	0.438	0.872	
Item 2	2.00	1.04	−0.123	−0.275	0.522	0.803	
Item 3	2.40	0.97	−0.088	−0.496	0.405	0.760	
Item 4	2.20	1.05	−0.208	−0.469	0.479	0.632	
Item 5	1.53	1.08	0.180	−0.740	0.703	0.520	
Item 6	1.17	1.19	0.774	−0.251	1.020		0.699
Item 7	0.64	0.98	1.557	1.952	1.538		0.627
Item 8	2.87	1.00	−0.724	0.150	0.348		0.518
Item 9	2.23	1.22	−0.327	−0.708	0.547		0.425

**Table 4 ijerph-19-10673-t004:** Factor loadings.

	Factor 1	Factor 2
Item 2	0.872	
Item 1	0.803	
Item 3	0.760	
Item 4	0.632	
Item 5	0.520	
Item 6		0.699
Item 7		0.627
Item 9		0.518
Item 8		0.425
Cronbach’s alpha	0.863	0.670

**Table 5 ijerph-19-10673-t005:** Psychometric properties for the Malay version of the SAVE-9 scale at the scale level (*n* = 203).

Methods	Psychometric Measure	Result	Suggested Cut-Off
CTT	Internal consistency measure using Cronbach’s alpha	0.830	>0.7
	Internal consistency measure using McDonald’s omega	0.827	>0.7
	Internal consistency measure using greatest lower bound	0.887	>0.7
	Test–retest reliability (Malay and original version)	0.921 **	See Note
	Convergent validity (SAVE-9 scale versus GAD-7 scale-Malay version)	0.530 **	See Note
	Convergent validity (SAVE-9 scale versus PHQ-9 scale-Malay version)	0.500 **	See Note

Note. ** Correlation is significant at the 0.01 level (two-tailed test); Correlation coefficients of <0.25 were considered as small; 0.25–0.50 as moderate; 0.50–0.75 as good; and >0.75 as excellent.

**Table 6 ijerph-19-10673-t006:** SAVE-9 Malay version, Factor 1, and Factor 2 scores among groups based on GAD-7 and PHQ-9 scores.

	Score of GAD-7 (Malay Version); Mean (SD)				Score of PHQ-9 (Malay Version) Mean (SD)			
	0 (*n* = 38)	1–4 (*n* = 67)	≥5 (*n* = 98)	Kruskal–Wallis Test Results	0 (*n* = 20)	1–9 (*n* = 135)	≥10 (n = 48)	Kruskal–Wallis Test Results
SAVE-9 total score	12.24 (5.45)	15.58 (4.28)	20.23 (5.95)	H = 49.638, *p* < 0.001	10.30 (5.10)	16.51 (5.01)	22.02 (6.13)	H = 48.738, *p* < 0.001
Factor 1 score	8.00 (4.15)	9.45 (2.79)	11.79 (3.97)	H = 27.797, *p* < 0.001	7.10 (3.40)	10.02 (3.38)	12.44 (4.51)	H = 25.196, *p* < 0.001
Factor 2 score	4.24 (2.41)	6.13 (2.43)	8.45 (3.05)	H = 54.320, *p* < 0.001	3.20 (2.02)	6.49 (2.62)	9.58 (2.96)	H = 59.792, *p* < 0.001

## Data Availability

The data presented in this study are available on request from the corresponding author.

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
