# Peer review of "Adaptation and Validation of the Malay Version of the SAVE-9 Viral Epidemic Anxiety Scale for Healthcare Workers"

_ijerph, 2022, doi:10.3390/ijerph191710673_

Round 1
Reviewer 1 Report
Thank you for the opportunity to review the manuscript. It is quite an interesting document, however I have some suggestions and observations to make:
The Introduction (pag 33-78) is good and emphasises that the questionnaire is applicable to healthcare workers, a professional category engaged in the front line during the COVID-19 pandemic. I would suggest the authors expand the references, in particular, the introduction does not refer to burnout, a widespread psychological phenomenon among healthcare workers during the COVID-19 pandemic (e.g., see: Borrelli et al. A new tool to assess burnout: the Italian version of the BAT for Italian healthcare workers. doi: 10.1186/s12889-022-12881-y).
Methods (pag 79-153) are well selected and the forward and backward translation, using WHO instrument validation protocols, is well explained. The sample size of the pre-test is well chosen with regard to age group, gender, marital status and healthcare workers. Validity Assessment and Reliability Assessment are ok.
Results and tabs (pag. 155-209) the results are well detailed, with additional tables to makes their understanding clear.
Discussion and Conclusion (pag. 210-256) is complete and encompasses all the points discussed above in a clear way.
Please, I would suggest expanding the references.
Author Response
Dear Examiner 1, we are grateful for your consideration of this manuscript, and we also very much appreciate your suggestions, which have been very helpful in improving the manuscript.
We have expanded the literatures by providing 5 additional references as suggested.
Reviewer 2 Report
The study is interesting and well conducted. The methodology is comprehensive and clear. The article could be published with the current manuscript but the manuscript's English level could be improved. This reviewer suggests English editing of the paper.
Author Response
Dear Examiner 2, we are grateful for your consideration of this manuscript, and we also very much appreciate your suggestions, which have been very helpful in improving the manuscript.
English has been improved and edited.
Reviewer 3 Report
The brief report "Adaptation and validation of the Malay version of the SAVE-9 viral epidemic anxiety scale for healthcare workers", as its title clearly shows, aims to adapt and validate the SAVE-9 scale to Malay.
The manuscript is well written and has been pleasant to read. However, I have some comments listed below:
1- Line 34: I guess “Humanity” is a tipo.
2- Please, add the cut-off of the Malay version of GAD-7 and PHQ-9 in Methods section.
3- Table 5: In the last line, you wrote “Convergent validity (Fear of Covid-19 scale versus PHQ-9 scale-Malay version)”. I guess you meant “Convergent validity (SAVE-9 scale versus PHQ-9 scale-Malay version)”
4- Line 220: Write “COVID-19” in capitols
Author Response
Dear Examiner 3, we are grateful for your consideration of this manuscript, and we also very much appreciate your suggestions, which have been very helpful in improving the manuscript. All the comments we received on this manuscript have been taken into account in improving the quality.
1- Line 34: I guess “Humanity” is a tipo.
- deleted.
2- Please, add the cut-off of the Malay version of GAD-7 and PHQ-9 in Methods section.
- cut-off values have been added in GAD-7 and PHQ-9 in Methods section.
3- Table 5: In the last line, you wrote “Convergent validity (Fear of Covid-19 scale versus PHQ-9 scale-Malay version)”. I guess you meant “Convergent validity (SAVE-9 scale versus PHQ-9 scale-Malay version)”
- changed to "SAVE-9 scale versus PHQ-9 scale-Malay version"
4- Line 220: Write “COVID-19” in capitols
- revised.
This manuscript is a resubmission of an earlier submission. The following is a list of the peer review reports and author responses from that submission.